# Gender-Related Biomechanical Properties of Masseter Muscle among Patients with Self-Assessment of Bruxism: A Comparative Study

**DOI:** 10.3390/jcm11030845

**Published:** 2022-02-05

**Authors:** Małgorzata Gałczyńska-Rusin, Małgorzata Pobudek-Radzikowska, Krzysztof Gawriołek, Agata Czajka-Jakubowska

**Affiliations:** Department of Orthodontics and Temporomandibular Disorders, University of Medical Sciences, 60-812 Poznań, Poland; malgorzata.pobudek@gmail.com (M.P.-R.); k.gawriolek@gmail.com (K.G.); agatacj@gmail.com (A.C.-J.)

**Keywords:** bruxism, muscle tone, muscle stiffness, muscle elasticity, masseter muscle

## Abstract

It seems extremely important to know the biomechanical properties of the orofacial tissues among patients with increased activity of masticatory muscles, such as bruxism. The aim of this study was to evaluate biomechanical properties of the masseter muscle by using MyotonPRO in adults with probable bruxism and to define gender relations. This study was conducted in the Temporomandibular Disorders Department at Poznan University of Medical Sciences, Poland (June 2021–November 2021) among patients that reported bruxism symptoms The patients underwent a clinical examination, then the biomechanical properties of the masseter muscles were assessed. The MyotonPro measured masseter tone, stiffness and elasticity in 36 patients with a self-assessment of bruxism (18 women and 18 men). Data were collected from relaxed and contracted muscles. In relaxed masseter muscles there were no statistically significant differences between the sexes in muscle tone, stiffness and elasticity. During contraction significant differences were found between the sexes in the tension and the stiffness of the masseter muscles. Moreover, women often experienced headaches in the temporal region and pain in the masseter muscles during palpation. Among patients with a self-assessment of bruxism, accompanying pain was significantly more frequent in women. Male gender was associated with increased muscle tension and stiffness of the contracted masseter muscle.

## 1. Introduction

The clinical signs of temporomandibular disorders (TMD) are commonly analyzed according to pain movement restrictions or increased muscles strain. However, there are a very limited number of papers describing the biomechanical factors of the masticatory muscles.

The masseter muscle is one of the most superficial and easy to examine muscles among the muscles of the stomatognathic system. Its main function is to elevate the mandible and exert chewing force and less in the protrusion and lateral movements related to chewing and swallowing or parafunctional activity [1]. It is known that some changes in muscle function or muscle strain can lead to many pathologies (myalgia and myofascial pain) as well as aesthetic changes in the appearance of the face [2,3,4]. Often, these disorders develop as a result of bruxism, which is defined as repetitive masticatory muscle activity, characterized by the clenching or grinding of teeth and/or strongly maintaining a certain position of the mandible or forcibly moving the mandible forward or to the side without contacting the teeth [5]. It is divided into sleep bruxism (SB) and awake bruxism (AB). Some papers divide it into possible bruxism when the diagnosis is based only on patients report, probable bruxism (based on clinical examination and symptoms) and confirmed, which is diagnosed by patient history, clinical examination and polysomnography (SB) or electromyography (AB) [3].

According to a systematic review by Melo et al. [6], the frequency of AB is 22–30% and SB is 1–15% in adults. The literature shows that [7] bruxism is not related to gender. Bruxism in adults is most common in people up to 40 years of age, and a significant decrease is observed after the age of 60 [7], which may be related to the loss of teeth with age. Manfredini et al. [8] suggest that a combination of psychosocial (stress and anxiety sensitivity), physiological/biological (e.g., neurotransmitters, such as dopamine and adrenaline, and genetic factors) and exogenous factors contribute to the risk of bruxism. The literature states that AB is associated mainly with the psychosomatic aspects, while SB is associated with a complex interaction of all factors influencing the functioning of the autonomic system [8]. The consumption of alcohol, caffeine, smoking, certain psychotropic drugs, gastroesophageal reflux disease, passive smoking, SB in childhood and genetic polymorphisms are also considered important risk factors for bruxism in adults [6,9].

The clinical symptoms characteristic of bruxism include the wearing of the teeth, painful tenderness of the masticatory muscles and TMJs, morning headaches, as well as fatigue and a feeling of tension in the stomatognathic system [8,10]. Moreover, bruxism may result in biomechanical complications associated with dental implants and implant-supported prostheses; however, the available evidence does not support adverse effects on other dental restorations or periodontitis [6].

Manfredini and Lobazzo [11] pointed to the relationship between bruxism, and the pain form of TMD. Melo et al. [6], in a systematic review of the literature, indicated a possible connection between bruxism and TMD. Jimenez-Silva et al. [12] suggested that bruxism may be associated with pain in the muscle fascia, and the pain and pathology in TMJ (displacement of discs and sounds in the joints).

Bruxism can also act as a protective factor, e.g., by increasing saliva secretion in reflux disease and, thus, increasing the protective effect against erosive acid action or the protection against obstructive sleep apnea by restoring the airway patency [6,13,14].

The masseter muscle can be diagnosed by the subjective assessment of the examiner (muscle hypergrowth) or objectively by the superficial electromyography (EMG), elastography (muscle stiffness) or by using a tool designed to assess muscle stiffness and elasticity, the MyotonPRO [4,15,16].

The aim of this study was to evaluate biomechanical properties of the masseter muscle by using MyotonPRO in adults with probable bruxism and to define the gender relations and reference values for bruxers.

## 2. Materials and Methods

This comparative study was conducted in the Temporomandibular Disorders Department at Poznan University of Medical Sciences, Poznan, Poland (June 2021–November 2021) among patients that reported bruxism symptoms.

### 2.1. Participants

Patients over 18 years of age who had a history of teeth grinding or were referred to the clinic due to suspected bruxism by the attending physician were qualified for the study. The exclusion criteria were predefined as follows: teeth loss, recent trauma, BMI > 30, muscle relaxant intake and patients over 40 years of age. Of the 45 patients enrolled in the study, five were disqualified due to missing teeth, three due to taking myorelaxants (Tolperisoni hydrochloridum) and one patient refused to undergo biomechanical muscle testing with MyotonPro. Thirty-six patients in total were investigated (18 women and 18 men).

### 2.2. Bruxism Diagnosis and Clinical Examination

The bruxism diagnosis was based on patient self-assessment. Patients were qualified into study by the self-assessment questionnaire according to the protocol presented by Lobbezoo et al [3,17]. A positive answer to one of the following questions was considered a confirmation of the presence of bruxism.

Have you been told, or did you notice yourself, that you grind your teeth or clench your jaws when you are asleep?Do you grind your teeth or clench your jaws during the day?

The clinical examination was performed using the Polish version of the Axis I Research Diagnostic Criteria for Temporomandibular Disorders Questionnaire (RDC/TMD) [18]. The clinical study also assessed the occurrence of possible symptoms of bruxism: masseter muscle hypertrophy, linea alba on the inside of the cheeks, pathological abrasion of the teeth, impressions on the tongue, as well as the occurrence of headache in the temple area in the last 30 days.

### 2.3. Instrumentation

The MyotonPro device (MyotonPRO, Myoton AS., Tallinn, Estonia) was used to evaluate the biomechanical properties of the muscles. The MyotonPRO provides a unique, reliable, accurate and sensitive way to objectively and non-invasively examine superficial skeletal muscles [16]. Performed with the MyotonPRO device, myotonometry is a non-invasive test that uses a very gentle mechanical impulse (0.4 N), applied to the skin, which allows the determination of the degree of deformation of the superficial tissues, detects the natural damping of soft tissue oscillations, as well as the oscillation frequency of the measuring element in the device, and on this basis it enables the determination of parameters such as the stiffness, tension and elasticity of the tissue [19]. A detailed description of the device has been presented in other publications [20]. In addition, Taş has proven that testing with the MyotonPRO device on the orofacial muscles is reproducible and reliable [21,22].

The following parameters of the masseter muscles were assessed using the MyotonPro:

F [Hz]: The oscillation frequency of the skeletal muscle, which characterizes the tone of the muscle (at rest) or the tension of the muscle (during contraction). During a contraction, the muscle tone increases.

S [N/m]: stiffness, which is the muscle’s ability to resist changes in form caused by an external force. During a contraction, the muscle stiffness increases

D: Logarithmic decrement, which characterizes muscle elasticity, is the ability of a muscle to restore its original shape after contraction. During a contraction, the muscle elasticity increases and the decrement decreases [19].

### 2.4. Procedure

During myotonometry, the patient laid on a dentist’s chair in a supine position. The most convex part of the muscle belly was selected for the study. The surface of the muscle (highest point) was marked with a washable marker during the maximum clenching of the teeth [23,24].

In the first part of the examination, patients were asked to relax their muscles without teeth contact. In the second part of the examination, they were asked to clench their teeth as much as possible. Both the right and left side muscles were examined. The testing end of the Myoton was placed on the skin surface perpendicular to the masseter muscle. The study was conducted by two dentists with many years of professional experience in diagnosing and treating patients with TMD.

### 2.5. Ethics Statement

The study was approved by the University Ethics Committee (consent number 522/21). The participants gave their written informed consent prior to participation in the study.

### 2.6. Statistical Analysis

Statistical calculations were conducted using SPSS v23. Software (IBM SPSS Statistics for Windows, Version 23.0. Armonk, NY, USA: IBM Corp). Descriptive statistics were used for determining the mean values, standard deviations (SD) and minimums and maximums of the demographic variables. The normality of the data distribution was checked with the Shapiro–Wilk test. For testing difference between independent groups, the *t*-test and Mann–Whitney U test were used. In all tests, a *p*-value < 0.05 was considered significant.

## 3. Results

The study involved 18 women and 18 men with self-reported bruxism. The mean age of the participants was 26 years (SD 6.0). The patient demographics are presented in Table 1.

The frequency of the possible symptoms of bruxism is presented in Table 2. The last line also includes the occurrence of pain during palpation of the masseter muscles, which is an integral part of the study according to the RDC/TMD questionnaire.

No statistically significant differences were found in the occurrence of the possible symptoms of self-reported bruxism between the sexes, except for the presence of headache in the area of the temples in the last 30 days preceding the examination and pain in the masseter muscles during palpation. Among women, these complaints occurred with a statistically higher frequency.

The biomechanical properties of the masseter muscles were first tested when the muscle was relaxed, without any contact between the upper and lower teeth (Table 3). There were no statistically significant differences between the sexes in any of the parameters.

In the next step, patients were asked to clench their teeth as much as possible, and the marked points of the masseter muscle were re-examined using the MyotonPro device. The results are presented in Table 4. Statistically significant differences were found between the sexes in the tension and stiffness of the masseter muscles during the maximum contraction.

According to the mechanical properties of the muscles, the oscillating frequency, stiffness and elasticity increased during contraction compared to the resting state of the masseter muscles. Elasticity is inversely proportional to the decrement; if the muscle decrement decreases, the elasticity of the muscle increases.

Both in the case of the biomechanical properties of relaxed muscles and in contraction, no statistically significant differences were found between the examined sides (right versus left, *p* > 0.05).

## 4. Discussion

There are relatively few studies using myotonometry involving the orofacial muscles [16,20,21,22,23,24,25,26,27,28]. Hence, there is a need to learn about the biomechanical properties of muscles, not only among healthy patients but also among people with various parafunctions, among people with malocclusion and among patients with TMD pain. For this reason, it was decided to study the biomechanical properties of the masseter muscles in people showing the parafunction of the masticatory system, such as grinding and clenching of the teeth. In our study, the frequency of possible tooth clenching and grinding symptoms did not differ between men and women, except for pain. Symptoms such as masseter muscle hypertrophy, linea alba, pathological tooth wear and tongue impressions were present in the vast majority of the studied patients. On the other hand, women with a self-assessment of bruxism experienced headaches in the temporal region and pain on palpation of the masseter muscles significantly more often than men, which is consistent with the observations of other authors [29].

Palpation is used in everyday clinical practice to assess the condition of the masticatory muscles. Manual palpation is the easiest way to measure muscle hardness. However, this subjective assessment depends on the experience and training of the examiner, and its measurement properties related to the assessment of masticatory muscle hardness are largely unknown [30]. Therefore, objective methods of assessing muscle tissues are sought.

In recent years, ultrasound imaging has been increasingly used to assess the elastic properties of tissues [4]. Elastography is a diagnostic method that consists of obtaining information about the cohesiveness of tissues and applying this information to morphological images, most often in the form of a transparent color map [30]. Strain elastography and shear wave elastography are used to assess the properties of the masseter muscles [31]. Strain elastography is based on the observation of tissue deformation under the influence of external mechanical stimuli. It allows the assessment of the hardness of the muscle. Shear wave elastography measures the speed of waves in the tissue caused by a high-frequency pulse and allows the assessment of the stiffness of the examined muscle. This method, unlike strain elastography, does not require the researcher’s experience in tissue compression and decompression [31]. Both methods require the use of specialized equipment and experience in the assessment of ultrasound images. According to Olchowy, shear wave elastography can be successfully used by dentists after appropriate training [4,32].

In our study, a MyotonPro device was used to assess the biomechanical properties of the muscles. It is intended for the examination of superficial skeletal muscles, tendons, ligaments and superficial myofascial tissues. MyotonPro, by the manufacturer’s recommendations, is not suitable for examining thin muscles that are less than 3 mm thick (e.g., temporal muscles) or for examining deep muscles covered with layers of other tissues (e.g., medial and lateral pterygoid muscles). Hence, for this study we selected the most accessible of all the masticatory muscles, the masseter muscle. The advantage of the device is its simplicity and short measurement time compared to other muscle tissue assessment techniques [33].

We did not find any gender-related differences at rest for any of the analyzed parameters. Ramazanoglu, Turhan and Usgu, in their study of a large group of patients without TMD, found significant differences in masseter elasticity between genders and some differences in muscle tension. As in our study, he found no difference in masseter muscle stiffness between women and men [23]. However, according to Dietsch et al., female sex is associated with greater stiffness of the orofacial tissues [20].

There is a small number of studies that evaluated the biomechanical properties of the masseter muscle during contraction. Yu, Chang and Zhang, with a group of healthy subjects, revealed an average masseter stiffness at rest of 369.5 N/m and with a contracted muscle of 618.3 N/m. In this research, masseter stiffness during contraction increased by 67.4% [16]. In our study stiffness increased by 77.4% on the right and by 76.3% on the left, accordingly. The differences in the obtained results can be explained by the muscle hyperactivity associated with clenching and the grinding of teeth occurring among the studied group. In patients with TMD, Lee and Chon found significant differences in masseter muscle stiffness and tension at the time of contraction compared to the control group, while finding no differences in muscle elasticity [25]. Song et al. found a significant difference in masseter stiffness during contraction between the healthy and paralyzed sides of the body in stroke survivors [26]. Another study that assessed masseter muscle stiffness in contraction is the Hara et al. study, but it evaluated elderly patients [27]. It did not find differences in masseter stiffness in the contraction state between the sexes. Our study showed significant differences between stiffness and muscle tension between men and women. In women, these values were significantly lower during muscle contraction. This can be related to the fact that the bite force reflecting the work of the masseter muscles in men is significantly higher [28]. It should be emphasized that our patients, compared to the Hara et al. subjects, belong to a much younger age group. It has been proven that age can have a significant impact on the biomechanical properties of muscles [20,34].

The above survey has its strengths and weaknesses. According to the authors’ knowledge, this is the first study in which myotonometry was used to assess the biomechanical properties of the masticatory muscles among bruxers. However, there are also some limitations. The diagnosis is based on probable bruxism, which is established on the basis of the patient’s questionnaire and clinical examination and does not require additional tests necessary to determine certain bruxism, such as polysomnography for diagnosing SB and EMG for diagnosing AB. Tooth wear, linea alba or impressions on the tongue can also occur in patients without bruxism, so these are not ideal indicators for diagnosing this parafunction [4]. It would be advisable to determine the biomechanical properties of other masticatory muscles as well, but at the moment these muscles are not available for study with the MyotomPRO device. The study did not compare the results to a control group that had not been diagnosed with bruxism, as this will be the subject of the authors’ next article

## 5. Conclusions

Among patients with a self-assessment of bruxism, accompanying pain was significantly more frequent in women. There was no difference between the sexes during muscle relaxation. Male gender was associated with increased muscle tension and stiffness of contracted masseter muscle.

## Figures and Tables

**Table 1 jcm-11-00845-t001:** Demographic characteristics of the participants.

Variable	Mean	Range
Age (in years)		
Men	26.9 (SD 6.5)	18–40
Woman	25.1 (SD 5.5)	19–40
BMI ^1^ (kg/m^2^)		
Men	24.3 (SD 3.1)	19.4–29.8
Women	20.6 (SD 2.2)	17–26.1

^1^ BMI, body mass index.

**Table 2 jcm-11-00845-t002:** The frequency of possible bruxism symptoms (percentages).

	All Individuals % (Present/Absent)	Men % (Present/Absent)	Women % (Present/Absent)	*p*-Value
Masseter hypertrophy	72.2/27.8	83.3/16.7	61.1/38.9	0.142
Linea alba	83.6/16.4	77.8/22.2	88.9/11.1	0.378
Tooth wear	61.1/38.9	61.1/38.9	55.6/44.4	0.739
Tongue impressions	83.6/16.4	88.9/11.1	77.8/22.2	0.378
Temporal headache	52.8/47.2	27.8/72.2	77.8/22.2	0.003 *
Masseter palpation pain	27.8/72.2	5.6/94.4	50/50	0.003 *

* Statistically significant differences (*p* < 0.05).

**Table 3 jcm-11-00845-t003:** Muscle tone, stiffness and elasticity of relaxed masseter muscles in all individuals and the distribution in terms of gender.

Masseter Relaxed	All Individuals	Men	Women	*p*-Value
Frequency (Hz)				
Right masseter	14.7 (SD 1.6)	15.0 (SD 2.0)	14.4 (SD 1.1)	0.264
Left masseter	14.5 SD (1.7)	14.8 (SD 1.8)	14.3 (SD 1.5)	0.318
Stiffness (N/m)				
Right masseter	294.2 (SD 47.7)	297.0 (SD 56.0)	291.3 (SD 39.2)	0.727
Left masseter	288.4 (SD 52.4)	288.1 (SD 45.0)	288.7 (SD 59.7)	0.973
Decrement				
Right masseter	1.8 (SD 0.2)	1.8 (SD 0.2)	1.8 (SD 0.3)	0.429
Left masseter	1.8 (SD 0.3)	1.7 (SD 0.2)	1.8 (SD 0.3)	0.418

**Table 4 jcm-11-00845-t004:** Muscle tension, stiffness and elasticity of contracted masseter muscle in all individuals and the distribution in terms of gender.

Masseter Contracted	All Individuals	Men	Women	*p*-Value
Frequency (Hz)				
Right masseter	20.3 SD (4.1)	21.8 SD (SD 3.8)	18.8 SD (4.0)	0.029 *
Left masseter	20.4 (SD 4.3)	22.3 SD (3.7)	18.5 SD (4.0)	0.006 *
Stiffness (N/m)				
Right masseter	521.9 (SD 187.4)	583.8 (SD 186.5)	460.1 (SD 171.5)	0.046 *
Left masseter	508.7 (SD 176.7)	587.8 (SD 160.1)	429.6 (SD 159.3)	0.005 *
Decrement				
Right masseter	1.4 SD (0.3)	1.3 (SD 0.3)	1.4 (SD 0.3)	0.341
Left masseter	1.3 (SD 0.3)	1.3 (SD 0.3)	1.4 (SD 0.3)	0.914

* Statistically significant differences (*p* < 0.05).

## Data Availability

The data presented in this study are available upon request from the author for correspondence. The data is not publicly available due to sensitive patient data.

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
