# Peer review of "Gender-Related Biomechanical Properties of Masseter Muscle among Patients with Self-Assessment of Bruxism: A Comparative Study"

_jcm, 2022, doi:10.3390/jcm11030845_

Round 1

Reviewer 1 Report

In this study, authors found that in the biomechanical properties of the masseter muscle, there is no sex difference at rest, but there is a sex difference during the exertion of maximum bite force. However, I think that the following information needs to be further discussed.

[Material and Methods]

1. Why did the authors exclude patients older than 40 years?

2. Patients who responded positively to either sleep bruxism (SB) or awake bruxism (AB) questions were recruited. The two types of bruxism are thought to differ in terms of the amount of muscle activity at the time of onset and duration. The symptoms in the oral-facial region caused by bruxism may also be different between SB and AB. Why did the authors assume only maximum clenching as the experimental condition?

3. RDC/TMD was used to screen for TMD. However, DC/TMD has been proposed in 2013. Why did you not use DC/TMD in this study? In DC/TMD, not only force but also duration of muscle palpation is clearly defined.

[Results]

1. For Tables 2, 3, and 4, enter the unit (N/m) in “Stiffness”.

2. Isn't " grind their teeth as much as possible " in line 151 a mistake for " clench their teeth as much as possible "?

3. Is it possible that the biomechanical properties of the masseter muscle are affected by the presence or absence of concomitant pain, in addition to gender differences?

[Conclusion]

1. It is indicated that the main purpose of this study is to determine the relationship between biomechanical properties of masseter muscle and gender. Do the conclusions need to specify the gender differences in the accompanying symptoms? We believe that the conclusion should state the results that are directly related to the main objective.

Author Response

Why did the authors exclude patients older than 40 years?

We excluded patients over 40 because there are indications that age also affects the biomechanical properties of muscles. Beyond this age, tooth deficiencies are much more common, which can also significantly impact the morphology of the masseter muscles.

The symptoms in the oral-facial region caused by bruxism may also be different between SB and AB. Why did the authors assume only maximum clenching as the experimental condition?

In our study, we assessed the biomechanical properties of the masseter muscle both at rest and in the state of maximum contraction that occurs when the teeth are clenched. We realize that AB and SB differ in essence. However, increased activity of the above-mentioned muscle occurs in both types of bruxism.

RDC/TMD was used to screen for TMD. However, DC/TMD has been proposed in 2013. Why did you not use DC/TMD in this study? In DC/TMD, not only force but also duration of muscle palpation is clearly defined.

We did not use the DC /TMD questionnaire due to the fact that the validation and translation of DC / TMD into Polish have not been published so far. We know that polish resarchers are working on this, and we look forward to the Polish version of the updated questionnaire.

Is it possible that the biomechanical properties of the masseter muscle are affected by the presence or absence of concomitant pain, in addition to gender differences?

We plan to conduct our research on a much larger group of patients, and then we will be able to answer the above question. However the literature review shows that there is probably no such relationship:

„The current evidence in favour of increased hardness in masticatory muscles in patients with myofascial TMD pain is weak, and the pathophysiological importance and clinical usefulness of such information remain unclear.”

Costa, Y.M.; Ariji, Y.; Ferreira, D.M. a. O.; Bonjardim, L.R.; Conti, P.C.R.; Ariji, E.; Svensson, P. Muscle Hardness and Masticatory Myofascial Pain: Assessment and Clinical Relevance. Journal of Oral Rehabilitation 2018, 45, 640–646, doi:10.1111/joor.12644.

All editorial comments have been made on the corrected text of the article.

We want to thank the Reviewer for taking the time and effort necessary to review the manuscript. We sincerely appreciate all valuable comments and suggestions, which helped us improve the manuscript's quality.

Reviewer 2 Report

This is an article focus on of an area with good justification and need to go deeper into the topic “Gender-related biomechanical properties of masseter muscle 2 among patients with self-assessment bruxism”. Overall most methods have been employed to a good standard and described well. I have a few comments and suggestions to help improve clarity in parts of the paper:

In the Title: It would be interesting to include the type of study.
In the Abstract: The abstract is very brief and more relevant information should be included:
-    It would be interesting to include the objective of the study clearly defined...The aim of the study is...to help better reading and understanding...
-    It would be interesting to include the selection criteria and what type of sampling was used.
-    It would be interesting to include the time period for data collection
-    It would be interesting to include the sampling used. How was the sample selected?
-    Include a period at the end of the abstract paragraph. 
In the Introduction:
-     The introduction section is very brief. More relevant information should be delved into, such as the prevalence of bruxism in men and women, factors that can cause bruxism and repercussions that bruxism can have on the temporomandibular joint, the entire stomatognathic complex and other associated repercussions such as headaches. 
In the Materials and Methods:
-    This section would need to be improved and structured in depth with subsections to facilitate reading and understanding (study design, ethical considerations, setting and participants, instruments, procedure, statistical analysis). In addition, it would be interesting to include a flow diagram of the participants and clearly specify the inclusion and exclusion criteria.
-    Include the type of study, how the participants were selected, specify whether it was by random sampling or not. 
-    Line 77: the number 36 should be thirty-six because it is the beginning of the sentence.
In the Discussion:
-    Line 168: Include a bibliographic citation of a study where myotonometry involving orofacial muscles is used.
In the conclusion: 
-    The concluding section is too brief, it would be interesting to also include the strengths and weaknesses of this study.

Author Response

In the Title: It would be interesting to include the type of study.  We add „comparative study” to the title

In the Abstract: The abstract is very brief and more relevant information should be included…

We tried to apply all the comments of the reviewer to the abstract, but we are limited by the journal's requirements as to the structure of the abstract: the abstract should be a total of about 200 words maximum.

In Introduction

We hope that we have comprehensively covered all comments from the reviewer

In the Methods

As recommended, we added subsections to facilitate reading and understanding.

In addition, it would be interesting to include a flow diagram of the participants and clearly specify the inclusion and exclusion criteria

In our opinion, there is no need to add a flow chart with such a small group of patients. Inclusion and exclusion criteria for patients are described in the subsection participants. If this is insufficient, please specify what exactly we would like to add.

Include the type of study, how the participants were selected, specify whether it was by random sampling or not

Among all patients reporting to the TMD Clinic in the period specified in the study, patients with bruxism were selected. In your opinion, should we enter convenience or purposive sample then?

In the conclusion: 
-    The concluding section is too brief, it would be interesting to also include the strengths and weaknesses of this study.

We described in the last paragraph of the discussion the limitations of this study. We have added one more sentence:

According to the authors' knowledge, this is the first study in which myotonometry was used to assess the biomechanical properties of the masticatory muscles among bruxists.

All editorial comments have been made on the corrected text of the article.

We want to thank the Reviewer for taking the time and effort necessary to review the manuscript. We sincerely appreciate all valuable comments and suggestions, which helped us improve the manuscript's quality.

Authors

Round 2

Reviewer 1 Report

Thank you for addressing the points I made.

Reviewer 2 Report

Dear Authors, 

This paper has improved a lot after making the suggested changes. Congratulation!

Best regards.